# The Pictorial Screening Memory Test (P-MIS) for Adults with Moderate Intellectual Disability and Alzheimer’s Disease

**DOI:** 10.3390/ijerph191710780

**Published:** 2022-08-30

**Authors:** Emili Rodríguez-Hidalgo, Javier García-Alba, Maria Buxó, Ramon Novell, Susana Esteba-Castillo

**Affiliations:** 1Specialized Service in Mental Health and Intellectual Disability, Institute of Health Assistance, Parc Hospitalari Martí i Julià, Catalonia, 17190 Girona, Spain; 2Research and Psychology in Education Department, Complutense University of Madrid, 28040 Madrid, Spain; 3Neurodevelopmental Group [Girona Biomedical Research Institute]-IDIBGI, Institute of Health Assistance (IAS), Parc Hospitalari Martí i Julià, Catalonia, 17190 Girona, Spain

**Keywords:** screening, memory, mild cognitive impairment, Alzheimer’s disease, dementia intellectual disability

## Abstract

In this study, we examined normative data and diagnostic accuracy of a pictorial screening test to detect memory impairment for mild cognitive impairment (MCI) and Alzheimer’s disease (AD) in Spanish-speaking adults with intellectual disability (ID). A total of 94 volunteers with ID (60 controls, 17 MCI, and 17 AD), were evaluated by neuropsychological tests including the PMIS-ID in a cross-sectional validation study. Discriminative validity between the MCI, AD, and control group was analyzed by the area under the ROC curve. A cut-off score of 4.5 on the immediate recall trial had a sensitivity of 69% and a specificity of 80% to detect memory impairment (AUC = 0.685; 95% CI = 0.506–0.863) in the AD group. The PMIS-ID is a useful screening test to rule out a diagnosis of memory decline in people with moderate level of ID and AD, and it shows good psychometric properties.

## 1. Introduction

Intellectual disability (ID) is a neurodevelopment disorder affecting intellectual functioning and adaptive behavior [1]. Globally, the prevalence of ID varies between 0.05 and 1.55% [2]. The underlying health conditions and the increased life expectancy of this population makes them more vulnerable to developing mild cognitive impairment (MCI) and dementia [3,4]. Although the epidemiological data of MCI and dementia in people with ID without Down’s syndrome (DS) has not been accurately set, recently, in Japanese people, the prevalence data for MCI hovers around 3% from the age of 45 onwards, and from 0.8% to 13.9% in those aged between 45 and 74 years old for dementia [5].

There is increasing evidence of a link between neurodevelopment disorders as ID and dementia [6]. Specifically, Alzheimer disease (AD) is the most common syndrome of dementia in DS [7], with incidence rates from 75% to 100% in those aged 60 and older [8]. Beyond DS, people with autism spectrum disorder (ASD) with ID are also at high risk for developing early onset AD (EOAD) [9], and ASD behaviors may be present in geriatric people with MCI and AD without ID [10]. Cerebral palsy (CP) by itself increases the risk of EOAD and related dementias [11], having an accelerated aging that predisposes patients to MCI and AD [12]. Based on these facts, certain ID conditions increase the risk of suffering AD. Therefore, early detection of MCI and AD in people with ID is required to implement appropriate interventions on the optimal therapeutic window.

It is well-known that identifying MCI or AD in people with ID poses a challenge to clinicians. Strengths and weaknesses of the cognitive and behavioral phenotype of each etiology must be taken into account when considering changes that may herald MCI or dementia in aging [13]. Therefore, some studies try to cognitively characterize the stages of MCI and AD in ID population. In people with mild or moderate levels of ID and unspecified etiology, decline in orientation questions and depressive symptoms [14] or decline in memory measured by a paired-learning task [15] have been found. By the Dementia Questionnaire for Mentally Retarded People [16], memory and orientation were altered [17]. Furthermore, cognitive decline is similar between people with ID and the general population with AD [18], exhibiting difficulties in learning and visuoverbal memory, semantic verbal fluency, and attention/executive functions, measured with the Fuld Object-Memory Evaluation [19], the Controlled Oral Word Association Test [20], and the Color Trail Test [21], respectively. In people with moderate and severe levels of ID, significantly lower scores were found in autobiographical memory and orientation in people with AD compared to healthy people [22]. In this same work, performance on a modified Objective Memory Test was lower in the group with AD, but only in the immediate recognition trial and in both immediate and delayed memory subtests of a Picture Recognition Test, a visuoverbal learning and memory test developed for the purpose of their work, that has proven able to detect memory troubles in people with moderate and severe levels of ID and DS [23]. Despite the different etiologies studied, memory decline arises as a common factor associated with MCI or AD.

As AD is the most common form of dementia and it is especially linked with DS, the majority of studies focused on this population have used cognitive measures. In this sense, a recent study of functional brain connectivity in people with DS [24] showed that low scores on the Cambridge Cognitive Examination for Older Adults with Down’s Syndrome (CAMCOG-DS) Spanish version [25] and decreased performance in verbal and visual memory appear to be key indicators of MCI and AD. Also, a slight impairment in delayed verbal and visual memory could be considered as a potential cognitive marker of MCI, with an increase in memory deficit in AD stage [26,27]. However, more important is the proposal model for MCI diagnosis in which decline or change in the Behaviour Regulaltion Index (BRI) from the Behaviour Rating Inventory of Executive Function-Informant’s Report (BRIEF) [28], and in the delayed verbal and visual memory domains from the Barcelona Test for Intellectual Disability (TB-DI) [29] are good predictors for MCI diagnosis [30]. Upon collecting these data, it is evident that memory declines across the AD continuum. Hence, there is a need to develop sensitive and screening test for daily clinical practice to detect memory changes associated with MCI or AD in this population.

Memory tests including a delayed recall trial are useful to detect longitudinal changes [31], as are recognition trials that improve the sensibility to detect retrieval alterations [32]. The word-based Free and Cued Selective Reminding Test (FCSRT) [33] for the general population and the pictorial Cued Recall Test (CRT) [34] for people with DS are tests with controlled learning techniques to optimize the coding processes [35] that are specially altered in the early stage of AD. It is well-known that scores of word and pictorial versions are not equivalent [36,37], although it is suggested that poor results in the free recall essay are associated with a reduction of the hippocampal volume in both tests [38]. As of today, a few screening tests are available that yield robust data with high accuracy in detecting memory deficits in the general population. For instance, the Memory Impairment Screen (MIS) [39] is a brief tool of four items that controls learning and cued recall, providing excellent properties to detect individuals at higher risk to develop AD. Different MIS-based versions are currently available, such as the Picture-Based Memory Impairment Screen (PMIS) [40], with four pictorial stimuli for English people with low levels of education. A Spanish pictorial memory test is also available that reports preliminary results for the general population with amnesic MCI and AD [41], as well as for people with DS [42]. When compared to the general population, there is no picture memory screening test available for Spanish people with ID. Furthermore, the administration of the available tests is hard to acquire for people with ID and time is short for routine appointments. Clearly, there is a need to develop valid memory screening tests suitable for primary care and specialized services in this population.

In light of these facts, the main aims of the current study were (a) to adapt the PMIS [40] for people with ID, (b) to provide normative data, and (c) to assess its feasibility as a screening tool for memory decline to discriminate healthy ID subjects from those with amnestic impairment typical of MCI or AD.

## 2. Materials and Methods

### 2.1. Participants

We undertook a prospective single-center cross-validation study with a convenience sample from December 2017 to December 2018. A total of 116 subjects recruited from the Specialized Service in Mental Health Unit for Adults with ID (SESM-DI, Parc Hospitalari Martí i Julià, Girona, Spain) were identified. All the participants were 18 years old or older, with a mild or moderate level of ID according to DSM-5 criteria (5th ed.; DSM-5) and with no drug treatment that could had significant effects on cognition. Those showing psychiatric or neurological conditions that could cause a dementia-like presentation or cognitive decline (depression, clinical hypo/hyperthyroidism, uncontrolled B9/B12 vitamin deficiency, seizures, delirium) and uncorrected auditory or visual sensory impairment that would make a neuropsychological assessment impossible were excluded. Of the subjects selected, 12 did not agree to participate in the study, 4 were excluded due to an absence of expressive language, and 6 more were excluded due to non-stable medical conditions at the moment of the assessment. The final sample consisted of 94 adults with ID (mean age = 47.33 ± 5.18 years; males = 57.4%, females = 42.6%) due to different etiology (54 Down’s syndrome, 4 tuberous sclerosis, 4 fragile X syndrome, 2 cerebral palsy, and 30 unknown). The baseline level of at least one year of all participants was available through annual follow-up in our service, in which most of the cognitive and functional tests that are part of the present study are routinely administered. The sample was divided into three groups: CN group (subjects without symptoms of MCI or AD), MCI group (subjects fulfilling criteria for MCI diagnosis), and AD group (subjects with AD diagnosis). The diagnosis of MCI or AD was based on expert multidisciplinary clinical judgment according to recent publications [27,30,43,44,45,46]. The diagnosis of MCI was made when participants presented a single or multiple cognitive decline(s) without significant functional loss. On the other hand, the diagnosis of AD was made in participants with memory decline and another cognitive impairment as aphasia, apraxia, agnosia, or disexecutive syndrome, and loss of functionality. In both conditions, changes from previous level of performance had to be supported by information obtained from a close caregiver [27,43,45].

The CN group included 60 subjects (47.47 ± 5.78 years; males = 65%, females = 25%) and was used to obtain normative data of the PMIS-ID. Both the MCI group, composed of 17 subjects (45.76 ± 4.22 years; male = 47.05%, females = 52.95%), and the AD group, composed of 17 subjects (48.41 ± 3.34 years; males = 41.18%, females = 58.52%), were used to gain the diagnostic accuracy properties of the PMIS-ID.

### 2.2. Instuments

Each participant underwent a comprehensive clinical and neuropsychological assessment. A neuropsychologist administered a large cognitive evaluation and informed-based measures during three different sessions.

#### 2.2.1. To Detect ID Level

Kaufman Brief Intelligence Test manual, second edition (K-BIT-2) [47];Vineland Adaptive Behaviour Scales-Second Edition (Vineland II) [48];

#### 2.2.2. Neuropsychological Assessment with Cognitive Tools Adapted and Validated for ID Spanish-Speaking Population

Barcelona Test for people with Intellectual Disability (TB-DI) [29]. This neuropsychological test battery consists of different subtests related to eight cognitive domains (language, working memory, orientation, praxis, attention, executive function, visuoconstruction, and memory). For this study, orientation and verbal learning were used (internal consistency of α = 0.87 and α = 0.73, respectively);CAMCOG-DS Spanish version [25]. This is the cognitive assessment module of the CAMDEX-DS Spanish version. It covers different cognitive domains mainly memory. For this study, memory subtests were used (new learning, remote, recent and memory total score).Picture Memory Impairment Screen for people with Intellectual Disability (PMIS-ID). It was applied at the beginning of the cognitive exam to mitigate the possible interferences with the rest of the tests.

#### 2.2.3. Parents Interview

Cambridge Examination for Mental Disorders of Older People with Down’s syndrome and Others with Intellectual Disabilities (CAMDEX-DS) Spanish version [25]. It consists of a structured informant-based interview, cognitive evaluation, diagnostic criteria guide, and recommendations for the interventions. The Spanish version presents an internal consistency of α = 0.93, considering performance on the memory subtest of the cognitive form and the memory section of the informant interview.

#### 2.2.4. Picture Memory Impairment Screen for People with Intellectual Disability (PMIS-ID)

The PMIS-ID consists of four-color photographs semantically unrelated in each quadrant of a DIN-A4 sheet. It includes four distinct parts: Identification (I), Learning (L), Immediate Recall (IR), and Delayed Recall (DR).

In the I and L parts, a sheet with four different categories of photographs (*horse*, *ludo*, *sofa*, *cherry*) is presented to the subject who has to name each one. Afterwards the subject has to identify them according to a cue (category) provided by the examiner (*animal*, *board game*, *fruit*, *furniture*). If the subject does not recognize or identify a photograph, the administration is ruled out. If items are correctly identified, the sheet is removed and the subject is told that they will be asked to repeat the words in a short time. Exploration must go on with another non visuoverbal task.

After three minutes, in the IR part, the participant is asked to recall the name of the four photographs (Immediate Free Recall, IFR). If any of the four items is missed, the examiner provides a category cue [Immediate Cued Recall (ICR)]. In case of failure, the target stimulus and two more distracters of an equal semantic category are orally provided (*cherry*, *pear*, *kiwi*; *goose game*, *cards*, *ludo*; *table*, *sofa*, *chair*; *horse*, *cow*, *tiger*) and the subject has to detect the right stimulus [Immediate Recognition (IRC)]. Correct stimulus has to be provided again if failure persisted. Finally, after twenty minutes, the DR part is administered, with the same tasks as in the IR part: Delayed Free Recall (DFR), Delayed Cued Recall (DCR) and a Delayed Recognition Recall (DRC).

The scores are calculated as two points for each correct response in FR, one point for each one in CR task, and 0.5 point in the RC. Immediate Total Recall (ITR) and Delayed Total Recall (DTR) are calculated separately (FR scores + CR scores + RC scores) of the FR, the CR and the RC parts. Total PMIS-ID (TPMIS-ID) score (0–16) is the sum of ITR and DTR, both ranging from 0 to 8.

PMIS-ID adaptation

The PMIS test [40] was adapted by: (1) introducing different items and categories suitable for people with mild and moderate ID; (2) translating the instructions with easiest vocabulary; (3) introducing Delayed Recall (DR) and Recognition (RC) tasks both for the Immediate Recall (IR) and Delayed Recall (DR) trials; and (4) by implementing a new scoring system. The numbers of items were consistent with the original version. An iterative procedure in line with practices recommended by Muñiz, Elosua, and Hambleton (2013) was followed, considering the particularity of this memory visuoverbal test. A pool of 12 color photographs belonging to four different semantic categories according to Spanish typicality norms were extracted [49,50]. To provide the sufficient complexity and avoid the ceiling effect, six stimuli corresponded to the first third while the other six to the second third of the total responses by category. The photographs were shown to 30 volunteers with mild and moderate ID (men age 43.8 ± 3.55; males = 55%; females = 45%) who were not enrolled in the validation study. Then, stimuli were reduced to the four most recognized and the remaining ones were introduced as distracters for the recognition task. Two native-English specialized psychologists in ID translated the test instructions. The two versions were discussed by the research team. An independent English linguist completed the back-translation of the document. Finally, a neurologist and a speech therapist reviewed the process and agreed to a pre-final version. The pre-final version was rounded off to implement further modifications during a pilot test in the same sample for the stimuli selection phase. During test administration, the examiner controlled the execution time and also checked the comprehension of the instructions, asking the volunteer to repeat and to explain them with their own words. The PMIS-ID was applied again in a convenience subsample of 20 participants within four weeks to assess test–retest reliability and in another 20 participants by two different examiners (SEC, ERH) to assess inter-rater reliability.

### 2.3. Data Analysis

The working database includes entries from February to December 2018. A descriptive analysis was applied to the entire group sample to describe the demographic variables (age and sex), the ID level, and performance on the cognitive protocol (TB-DI memory and orientation, CAMCOG-DS memory subtest). Due to the small sample size in some subgroups, a nonparametric statistical analysis was conducted. Means comparisons were made by independent samples t-test or ANOVA for qualitative data, and χ^2^-test for category data. These data were presented as median and interquartile range (IQR) and were compared by Kruskal–Wallis test with Bonferroni adjustment. Multiple linear regression analysis was used to verify the possible influence of sociodemographic variables (age, sex) and the level of ID (mild, moderate) on the PMIS-ID immediate total score, delayed total score, and total score. Reliability was estimated by the test–retest and inter-rater methods and by calculating Pearson and intraclass correlation coefficients, respectively. For the MCI and AD groups, a descriptive analysis was applied for the sociodemographic variables (age, sex, and ID level). Construct validity of the PMIS-ID was verified using the coefficient corrected kappa statistic between the PMIS-ID total score and MCI and AD groups. Spearman’s rho was run to evaluate convergent validity between the PMIS-ID total score and the memory subtest performance of the TB-DI and the CAMCOG-DS. Normative data was presented in line of the assumption of the MCI or AD prevalence (%) for the different PMIS-ID cut-scores (PPV and NPV). Receiver operating characteristic curve (ROC) and the Youden index were used to determine the optimal cut-off point of the PMIS-ID (immediate, delayed, and total scores) as a screening memory test for MCI or AD. The areas under the curve (AUC) were compared between the different trials [51].

All statistical analyses were conducted using the software program G-Stat (version 2.0) and the statistical software program SPSS (version 27.0; SPSS Inc., Chicago, IL, USA). Bilateral significance levels were set at a *p*-value of less than 0.05.

## 3. Results

### 3.1. Demographics

Results in Table 1 shows that the CN group was similar to the MCI and AD groups in mean age (*p* = 0.315), level of ID (*p* = 0.159), and sex distribution (*p* = 0.136).

### 3.2. Between-Group Comparison of Cognitive Performance

Compared to the CN group, the scores were significantly lower for the AD group on verbal learning, delayed free recall subtest, and false positive scores of the TB-DI and on new learning, remote, recent, and memory total score subtest of the CAMCOG-DS. Performance on the subscales of both tests was similar between the MCI and CN groups (Table 1).

In people with mild ID, the performance of the CN and MCI groups was similar on the entire subtest. Between the CN group and AD group, significant differences were shown in performance on verbal leaning, delayed free recall, and false positives of the TB-DI. No significant differences were observed on the subtests of the CAMCOG-DS (Table 2).

In people with moderate ID, no significant differences were observed between the CN group and the MCI group on all the subtests. Significant differences were observed on the false positives score, and no significant differences but moderate decline were observed on verbal learning and delayed free recall subtests of the TB-DI. Significant differences were observed on new learning, remote, recent, and memory total score of the CAMCOG-DS (Table 3). 

### 3.3. Between-Group Comparison of PMIS-ID Performance

The AD group performance differs from the CN group on free recall and total scores from both immediate and delayed part, as well as on the total PMIS-ID score, whereas the performance of the MCI and CN groups was similar on all the trials (Table 4). Analyses according the ID level show that these differences were valid for adults with a moderate level of ID, but not for those with mild ID (Table 5).

### 3.4. Analysis of Control Group

To study the demographics and level of ID impact on the PMIS-ID scores, a multiple linear regression analysis for the PMIS-ID immediate total score (*r2adj* = 0.032; *F* = 1.006; *p* = 0.397), delayed total score (*r2adj* = 0.515; *F* = 1.102; *p* = 0.356), and total score (*r2adj* = 1.357; *F* = 1.271; *p* = 0.293) was performed for the CN group. Neither the level of ID (immediate: *t* = −1.523, *p* = 0.133, *B* = −0.615; delayed: *t* = −1.419, *p* = 0.161, *B* = −0.895; total: *t* = −1.509, *p* = 0.069, *B* = −1.509) nor the age (immediate: *t* = −1.078, *p* = 0.286, *B* = 0.0.38; delayed: *t* = −0.689, *p* = 0.494, *B* = −0.037; total: *t* = −0.001, *p* = 0.999, *B* = −0.001) nor sex (immediate: *t* = −0.602, *p* = 0.549, *B* = −0.615; delayed: *t* = −0.350, *p* = 0.7274, *B* = −0.222; total: *t* = −0.568, *p* = 0.572, *B* = −0.466) were significantly related to the PMIS-ID total scores. Hence, no adjustment of the PMIS-ID total scores was required.

### 3.5. Reliability

The Pearson’s correlation coefficient for the total PMIS-ID score was 0.90 (*p* < 0.0001), a very strong test–retest association. The inter-examiner agreement was found to be good, with a mean intra-class correlation coefficient (ICC) of 0.96 (*p* < 0.0001).

### 3.6. Validity

#### 3.6.1. Convergent Validity

The correlations patterns between the PMIS-ID total score for the whole sample and by level of ID and the TB-DI and CAMCOG-DS subtest are displayed in Table 6.

Considering the total sample, the overall correlations were significant. For the mild ID sample, the total score of the PMIS-ID was low but positive in correlation with the verbal learning and delayed free recall subtest of the TB-DI and with the new learning ant total memory score of the CAMCOG-DS. In the moderate sample, satisfactory positive correlations were found between verbal learning and delayed free recall subtest of the TB-DI, and new learning, recent, and total memory score subtest of the CAMCOG-DS. Low negative correlation was found between false positives score of the TB-DI and total PMIS-ID total score. Weak positive and negative correlations were found for the other subtest of both sample and the PMIS-ID total score.

#### 3.6.2. Discriminative Validity

Three ROC curves for each total PMIS-ID trial with enough discriminatory power between CN and AD groups with moderate ID were plotted (Figure 1) and comparisons between AUCs were calculated (Appendix A of the Online Supplement). The AUCs to discriminate between CN and AD groups had good diagnostic utility. For immediate total recall (ITR), the AUCs were 0.685 (CI 95%: 0.506–0.863) and 0.757 (CI 95%: 0.612–0.903) for the total delayed recall (TDR) and 0.749 (CI 95%: 0.597–0.901) for the TPMIS-ID. Paired comparison of the AUCs of the three totals scores were not statistically significant (*z* = −0.842, *z* = −1.021, *z* = 0.229). Hence, immediate total recall (ITR) was the best and most time-efficient trial for our purpose.

### 3.7. Normative Data

The values of sensitivity, specificity, positive likelihood ratio (+ LR), negative likelihood ratio (−LR), and the Youden indexes for various cuts-off scores of the PMIS-ID for the AD group with mild and moderate level of ID were calculated. For the group with moderate level of ID, the delayed and total recall trials normative data are presented in Appendix A of the Online Supplement, and for the immediate recall (IR) in Table 7. Data for the sample with mild ID and AD is presented in Appendix A of the Online Supplement. Considering that the base rate in the moderate ID sample was 25.5%, for the immediate total recall (ITR), a cut-off score of ≤4.5 provided a moderate sensitivity (69%) and a high level of specificity (80%).

## 4. Discussion

This is the first validation study of a pictorial screening memory test for people with ID, providing normative data and assessing its diagnostic utility to detect memory decline. The PMIS-ID demonstrates good discriminant validity for distinguishing between people with moderate ID and AD from healthy population with ID, and exhibits good convergence validity and reliability.

Our results show that it is possible to detect memory impairment with the PMIS-ID in people with moderate ID in a quick and simple way. It is easy to administer, brief (no more than five minutes), and cost-effective. Also, it shows good convergent validity with memory subtest of the TB-DI and CAMCOG-DS. Considering that the verbal memory subtest of the TB-DI presents good internal consistency [29] and the CAMCOG-DS is recommended for follow-up studies [52], this proves that the PMIS-ID measures memory processes. Furthermore, AUC is acceptable (above 0.7) and the discriminant validity for the proposed cut-off score of the PMIS-ID (4.5) shows good specificity (86%) and appropriate sensitivity (69%). Values of specificity are in line with the majority of the MIS-based screening tests in the general population that yield specificities values higher than 80%: the MIS [39], MIS-S [53], MIS-E [54], MIS-D [55], and PMIS [40]. The PMIS-ID also identifies healthy subjects correctly as the aforementioned screening tests. Consequently, the PMIS-ID is an excellent screening memory test for use in daily clinical specialized services for people with moderate ID.

Data gained from the immediate total recall (ITR) part of the PMIS-ID in isolation well enable detection of memory impairment in adults with moderate ID and AD. The inclusion of a delayed recall is tautological, opposite to the improvement described in the general population with MCI [53], but closer to the use of pictorial memory test based in learning trials without delayed recall in people with low educational level [41] and in people with ID [42]. Our results reveal that when performance is scarce in the immediate recall, it is also in the delayed recall. Moreover, in our sample, people with lower performance in free recall do not gain with the inclusion of a cued recall in the immediate or in the delayed recall. These results are aligned with those described for the MIS-S [54] and cued recall test [42]. Also, learning of relational material depends on the functionality of the hippocampus [55] and free recall trials of the FCSRT picture versions could be considered as an indicator of hippocampus structural integrity [38]. Furthermore, longitudinal memory score variation is specifically associated with volume change in the hippocampus [45,56]. For these reasons, it can be concluded that the PMIS-ID is a noninvasive tool to detect memory impairment due to hippocampus dysfunction in people with moderate ID and AD.

In our study, significant memory decline is the predominant symptom in people with moderate level of ID and AD. Memory decline has been described in MCI and AD in the general population and those with ID [57]. Especially, in people with DS, a slight memory impairment is usually found in MCI, with a significant decline in AD [24,26,27,30,58]. Possibly, memory processes in people with moderate level of ID are not semantic-dependent, and other strategies should be considered because deep processing is beyond their capability [59]. Other promising alternatives could involve developing a test under the associative learning paradigm (binding) that depends on the integrity of the medial temporal lobe structures, as in the general population with hopeful results for AD [60], MCI [41], and in adults with mild and moderate ID [61].

Although stimuli selection in our work was accurate, the scores on the PMIS-ID have a low ceiling effect. Despite the apparent easiness of the four proposed pictorial stimuli, the PMIS-ID measures the same function as a standard memory test. High concordance was obtained between the PMIS-ID total score and memory subtests of the CAMCOG-DS [25] and the TB-DI [29], especially for people with a moderate level of ID. For instance, the satisfactory concordance in our study for the verbal learning subtest of the TB-DI (*r* = 0.62, *p <* 0.001) has also been described between the MIS-E [62] and the analogue subtest of the original version of the TB-DI, the Barcelona Test [63] (*r* = 0.78; *p <* 0.001) in the general population. In this sense, it is important to consider that assessing memory in people with ID is a complex task for clinicians. In clinical practice, pictorial tests seem to be better accepted due to an increased feeling ability to solve the task. In return, the ceiling effect in pictorial tests has already been considered in the general population because the selected drawings are excessively simple and with little ecological validity [64]. This is consistent with the enhancement of performance with pictorial memory test in older adults [37] and in a Spanish population with amnesic MCI [41] and with AD [36] compared with word memory test. Our results confirm the classical dual-coding information theory [65] that postulates superiority in picture processing against words. In this context, increasing the number of stimuli could reduce the ceiling effect, improving its diagnostic capacity for adults with mild ID.

Our results show a slight decrease of the performance on the PMIS-ID for MCI compared with the CN group, both for mild and moderate level of ID groups. Medians are similar in both groups for each trial, but values of the first and third quartile are wider again in each trial, especially for the moderate ID group. This fits with the discrete accuracy to detect MCI compared with the satisfactory data for AD in people with ID. Until recently, few studies had shed light on how to detect preclinical or prodromal stages of AD. Thus, in the recent proposed diagnostic criteria for MCI in DS [30], three variables from a comprehensive neuropsychological examination have proven sensitive enough for this fact: the BRI of the BRIEF and the abstraction and delay memory subtest from the TB-DI. Also, the PAL first-trial memory is one of the most sensitive variables to detecting changes between the preclinical and prodromal phases of AD in people with DS [7]. Furthermore, it is feasible to diagnose AD with neuropsychological tools such as the CAMCOG-DS or the modified Cued Recall Test (mCRT) [34]. In this sense, a decline of performance on both tests was evident in the continuum of AD, but performance on the mCRT was not discriminative for people with DS in the prodromal stage of AD. Analyzing these results, one can deduce that the most tangible changes occur when AD is established, but not in the prodromal phases. Possibly, the course of AD is different in comparison to the general population, in which MCI can be considered a slowly progressive transitional phase in cases of conversion to AD; sensitive neuropsychological instruments are available to detect these changes. Based on results in subjects with ID, changes between the preclinical and prodromal phase would be minimal and undetectable with current neuropsychological instruments, and are only sensitive when performance declines abruptly. From this data, it can be hypothesized that memory deficits with more or less intensity are part of the phenotype of almost all people with ID, and this implies that falls in memory are more difficult to detect than in the general population, in which the margins of scores are greater. This also might suggest that the course of the disease may not be slowly progressive and early detection is essential to initiate proper intervention. Therefore, neuropsychological tools with normative data for all ID ranges are needed to reduce misdiagnosis and to interpret cognitive profiles better in normal and pathological ageing.

Overall, our clinical experience reveals that in the general population, the diagnostic of MCI or DA poses a challenge in those with high educational level, just as learning disorders can be masked in exceptional children. In this sense, intelligence, education, and occupational level influence the onset and course of deterioration due to the reserve cognitive assumption [66]. This is also evident in people with ID in which the presentation and natural history of AD varies according to the level of ID [46]. Furthermore, declines in CAMCOG-DS scores are more evident in people with moderate ID [56], and adults with Klinefelter syndrome with higher values in intelligence tests performed better in working memory and executive functions [67]. Thus, the impact of the level of intelligence on neuropsychological tool performance seems contrasted. That is why sufficiently reliable neuropsychological tools should be available for different ranges of intellectual capacity and the level of cognitive reserve must be regarded.

There are some limitations that need to be considered when interpreting our findings; some caution is required.

First, even if the sample is acceptable for a preliminary study, participants were classified by level of ID, decreasing the size of the MCI and AD groups and limiting statistical and discriminatory power. Therefore, future research should be carried out with a representative sample, calculating power estimation before the onset of the study.

Second, alternatives forms are desirable to use in clinical practice and research in neuropsychology. In our study, the PMIS-ID stimuli were chosen according to the data in the general population, which could contribute to the ceiling effect of the scores. To avoid this bias and to expand its use for people with mild ID, we recommend carrying out studies to provide norms for word prototipicity and picture familiarity, according to appropriate cultural context, for people with ID.

Third, the predictive capacity of the PMIS-ID cannot be evaluated reliably because it is a cross-sectional study. Also, we have not considered the stage or severity degree of AD. Our objective was to develop a rapid measure of memory decline associated with MCI or AD to be applied in daily primary care, but it is well-known that follow-up of cognitive decline is required in people with ID to confirm a diagnosis of MCI or AD. Therefore, we recommend carrying out longitudinal studies with various time points and adapting current staging scales in the general population for people with ID.

## 5. Conclusions

Limitations notwithstanding, the PMIS-ID is a valid memory screening test for people with a moderate level of ID for use in primary health care centers and in clinical specialized services. The findings in the current pilot study suggest that the PMIS-ID does not provide a comprehensive memory assessment but may be useful as a first step in the diagnostic process to help clinicians in healthcare settings to determine the need to carry out a broader diagnostic evaluation.

## Figures and Tables

**Figure 1 ijerph-19-10780-f001:**
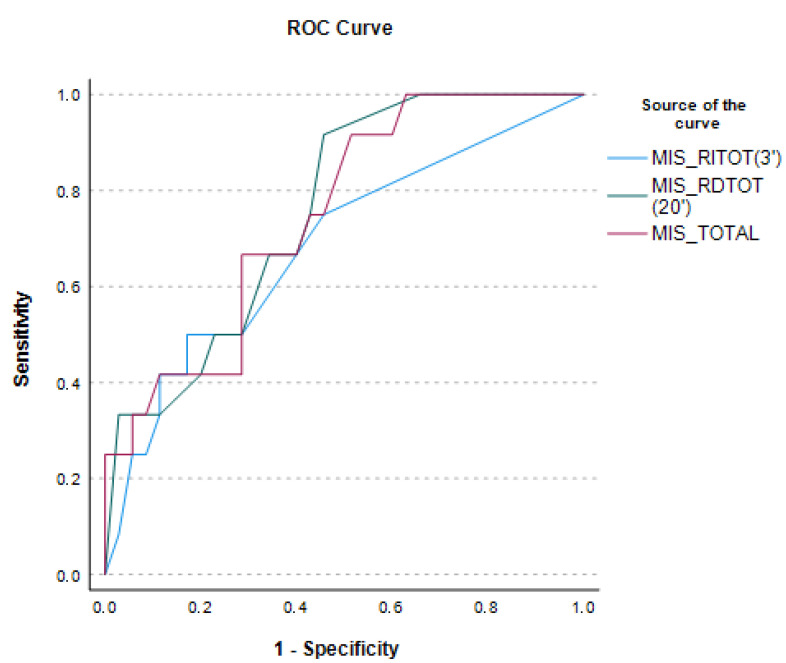
Receiver operating characteristics (ROC) of the PMIS-ID as a screening memory tool for AD in people with a moderate level of ID.

**Table 1 ijerph-19-10780-t001:** Demographic characteristics and cognitive performance by group.

	Control	MCI	AD
*n*	60	17	17
Age	47.47 ± 5.78 (40–60)	45.76 ± 4.22 (42–55)	48.41 ± 3.34 (43–54)
SEX			
Male	39 (65%)	8 (47.06%)	7 (41.18%)
Female	21 (35%)	9 (52.94%)	10 (58.82%)
ID LEVEL			
Mild	32 (46.67%)	10 (58.82%)	5 (29.41%)
Moderate	28 (53.33%)	7 (41.18%)	12 (70.59%)
TB-DI			
Verbal learning	26.7 (20–33.5)	24.4 (16–31)	17.6 (9–24) ***
Delayed free recall	4.7 (2.5–7)	3.6 (0–6)	1.6 (0–3) ***
False positives	3.5 (0–6.5)	6.1 (1–12) *	8.5 (4–12) ***
Delayed word recognition	10.2 (10–12)	11.4 (12–12)	10.6 (10–12)
CAMCOG-DS			
New learning	13.5 (11–16)	11.5 (8–15)	8.8 (7–11) ***
Remote	2.7 (2–4)	2.6 (2–4)	1.5 (0–2) ***
Recent	2.6 (2–4)	1.9 (1–3)	0.9 (0–2) ***
Memory total	18.8 (15–23)	16.1 (11–21)	11.2 (9–15) ***

Values are given by means and range; sex in percentages for each group. ID, intellectual disability; TB-DI, Barcelona Test for People with Intellectual Disability; CAMCOG-DS, Cambridge Examination for Mental Disorders of Older People with Down’s Syndrome and Others with Intellectual Disability (brief neuropsychological battery); MCI, mild cognitive impairment; AD, Alzheimer’s disease. * *p* < 0.05, *** *p* < 0.001, using Kruskal–Wallis with Bonferroni correction or χ^2^-test for category data between MCI and AD groups compared with the control group.

**Table 2 ijerph-19-10780-t002:** Cognitive scores by group for mild intellectual disability sample.

	CN	MCI	AD
*n*	32	10	5
TB-DI			
Verbal learning	31.2 (26–36)	26 (17–33)	20 (17–24) **
Delayed free recall	5.9 (4–8)	3.8 (0–7)	1.8 (1–3) **
False positives	2.2 (0–3)	4.2 (1–9)	6.8 (3–12) *
Delayed word recognition	10.5 (10–12)	10.9 (9–12)	11.2 (10–12)
CAMCOG-DS			
New learning	14.2 (12–16)	12.6 (9–15)	11.8 (10–11)
Remote	2.9 (2–4)	2.9 (2–4)	2.2 (2–3)
Recent	2.9 (2–4)	2.2 (1–4)	1.4 (0–2)
Memory total	19.9 (17–23)	17.7 (13–21)	15.4 (11–19)

Values are given by means and range. TB-DI, Barcelona Test for People with Intellectual Disability; CAMCOG-DS, Cambridge Examination for Mental Disorders of Older People with Down’s Syndrome and Others with Intellectual Disability (brief neuropsychological battery); CN, control group; MCI, mild cognitive impairment; AD, Alzheimer’s disease. * *p* < 0.05, ** *p* < 0.01, using Kruskal–Wallis with Bonferroni correction between MCI and AD groups compared with the control group.

**Table 3 ijerph-19-10780-t003:** Cognitive scores by group for moderate intellectual disability sample.

	CN	MCI	AD
*n*	28	7	12
TB-DI			
Verbal learning	21.5 (17–26)	22.1 (15–24)	16.6 (4.5–24)
Delayed free recall	3.3 (1–4)	3.4 (0–6)	1.6 (0–2)
False positives	4.9 (0–10)	9.3 (6–12)	8.9 (3–12) **
Delayed word recognition	9.9 (7.5–12)	12 (12–12)	10.4 (10–12)
CAMCOG-DS			
New learning	12.8 (11–15)	10 (5–13)	7.6 (4.5–11) ***
Remote	2.6 (2–4)	2.3 (2–3)	1.3 (0–2) **
Recent	2.3 (0.5–4)	1.6 (0–2)	0.7 (0–2) **
Memory total	17.5 (15–21)	13.9 (9–18)	9.5 (5.5–13) ***

Values are given by means and range. TB-DI, Barcelona Test for People with Intellectual Disability; CAMCOG-DS, Cambridge Examination for Mental Disorders of Older People with Down’s Syndrome and Others with Intellectual Disability (brief neuropsychological battery); CN, control group; MCI, mild cognitive impairment; AD, Alzheimer’s disease. ** *p* < 0.01, *** *p* < 0.001, using Kruskal–Wallis with Bonferroni correction between MCI and AD groups compared with the control group.

**Table 4 ijerph-19-10780-t004:** PMIS-ID descriptive scores by group.

	CN	MCI	AD
*n*	60	17	17
*Immediate*			
Free recall	4 (3–4)	3 (2–4)	2 (0–4) **^a^
Cued recall	0 (0–1)	0 (0–1)	0 (0–1)
Recognition	0 (0–0)	0 (0–0)	0 (0–1)
Total	8 (7–8)	7 (6–8)	6 (2.5–8) **^a^
*Delayed*			
Free recall	4 (3–4)	3 (0–4) *^b^	0 (0–3) ***^a^
Cued recall	0 (0–1)	0 (0–1)	1 (0–2)
Recognition	0 (0–0)	0 (0–1)	0 (0–1)
Total	7.5 (6.5–8)	7 (2.5–8)	3.5 (1–6.5) ***^a^
*Total PMIS-ID*	15 (11.8–16)	13 (9.5–16)	10.5 (4–13) **^a^

Median (first quartile–third quartile) for each variable is summarized. PMIS-ID, Picture Memory Impairment Screen for People with Intellectual Disability; CN, control group; MCI, mild cognitive impairment; AD, Alzheimer’s disease; ^a^ Between CN and AD group. ^b^ Between MCI and AD group. * *p* < 0.05, ** *p* < 0.01, *** *p* < 0.001 using Kruskal–Wallis with Bonferroni correction between MCI and AD groups compared with the control group.

**Table 5 ijerph-19-10780-t005:** PMIS-ID descriptive scores by group and level of intellectual disability.

	Mild ID			Moderate ID		
	CN	MCI	AD	CN	MCI	AD
*n*	32	10	5	28	7	12
*Immediate*						
Free recall	4 (3–4)	4 (2–4)	2 (1–4)	4 (3–4)	3 (0–4)	1.8 (0–3.5)
Cued recall	0 (0–1)	0 (0–1)	1 (0–2)	0 (0–1)	0 (0–1)	0 (0–1)
Recognition	0 (0–0)	0 (0–0)	0 (0–0)	0 (0–0)	0 (0–1)	0 (0–1)
Total	8 (7–8)	8 (6–8)	6 (3.5–8)	8 (7–8)	6.5 (0.5–8)	5.2 (1.3–7.5) *^ab^
*Delayed*						
Free recall	4 (3–4)	4 (2–4)	2 (1–4)	3 (2–4)	0 (0–3)	0 (0–2.5) **^a^
Cued recall	0 (0–1)	0 (0–1)	1 (0–1)	0 (0–1)	1 (0–2)	0.5 (0–2)
Recognition	0 (0–0)	0 (0–0)	0 (0–0)	0 (0–0)	0 (0–1.5)	0.3 (0–1)
Total	8 (7–8)	8 (6–8)	5 (3–8)	7 (3.5–8)	2.5 (1–7)	0.3 (0–6.3) **^a^
*Total PMIS-ID*	16 (14–16)	15.5 (12–16)	11 (6.5–16)	14 (11–16)	10 (1–13.5)	10.5 (1.3–12.3) **^a^

Median (first quartile–third quartile) for each variable is summarized. PMIS-ID, Picture Memory Impairment Screen for People with Intellectual Disability; ID, intellectual disability; CN, control group; MCI, mild cognitive impairment; AD, Alzheimer’s disease. ^a^ Between CN and AD group. ^b^ Between MCI and AD group. * *p* < 0.05, ** *p* < 0.01, using Kruskal–Wallis with Bonferroni correction between MCI and AD groups compared with the control group.

**Table 6 ijerph-19-10780-t006:** Convergent validity values by groups.

	PMIS-ID Total Score		
	Total	Mild ID	Moderate ID
TB-DI			
Verbal learning	0.622 **	0.409 **	0.665 **
Delayed free recall	0.549 **	0.381 **	0.558 **
False positives	−0.459 **	−0.216	−0.522 **
Delayed word recognition	−0.215 *	−0.138	−0.289 *
CAMCOG-DS			
New learning	0.688 **	0.519 **	0.763 **
Remote	0.395 **	0.182	0.398 *
Recent	0.454 **	0.186	0.560 **
Total memory	0.679 **	0.481 **	0.743 **

TB-DI, Barcelona Test for People with Intellectual Disability; CAMCOG-DS, Cambridge Cognitive Examination Adapted for Individuals with Down Syndrome; PMIS-ID, Picture Memory Impairment Screen for People with Intellectual Disability; ID, intellectual disability. * *p* < 0.05, ** *p* < 0.01 (two-tailed test), using Spearman’s correlation coefficient.

**Table 7 ijerph-19-10780-t007:** Normative data of the PMIS-ID immediate total score for different cut-off points for AD in moderate level of ID sample.

Cut-Off Points	S	Sp	*J*	PPV ^a^	NPV ^a^
0	0.00	1.00	0.00	0.64	1.00
0.5	0.31	1.00	0.31	0.67	1.00
1	0.50	1.00	0.50	0.71	1.00
1.5	0.50	1.00	0.50	0.72	1.00
2	0.50	0.86	0.36	0.72	0.94
2.5	0.50	0.96	0.46	0.74	0.94
3	0.56	0.96	0.52	0.76	0.94
3.5	0.56	0.90	0.46	0.74	0.84
4	0.62	0.90	0.52	0.76	0.84
4.5	0.69	0.86	0.55	0.76	0.80
5.5	0.69	0.83	0.52	0.75	0.76
6.5	0.75	0.79	0.54	0.74	0.73
7.5	0.87	0.62	0.49	0.78	0.59
8	1	0.00	0.00	1.00	0.36

MIS-ID, Picture Memory Impairment Screen for People with Intellectual Disability; S, sensitivity; Sp, specificity; *J*, Youden’s J statistic; PPV, positive predictive values; NPV, negative predictive values. ^a^ 25.5% is the prevalence of AD in the sample.

## Data Availability

The data presented in this study are available on request from the corresponding author.

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
