# Peer review of "The Pictorial Screening Memory Test (P-MIS) for Adults with Moderate Intellectual Disability and Alzheimer’s Disease"

_ijerph, 2022, doi:10.3390/ijerph191710780_

Round 1

Reviewer 1 Report

The authors conducted an observational study to examine the accuracy of the Picture Memory Impairment Screen for People with Intellectual Disability (PMIS-ID) in detecting memory impairment in Spanish-speaking adults with intellectual disability (ID). A total of 94 adults with ID and either AD, MCI, or normal cognition were recruited and received PMIS-ID and standard clinical and neuropsychological assessment. They showed that PMIS-ID was valid and reliable. In addition, among those with moderate ID, it accurately discriminates those with AD from those without MCI and AD. These works are essential in light of the urgent need to develop a practical screening tool for memory impairment in adults with ID.

There are some comments.

Comments:

1.      Methods (Participants, Line 108 on Page 3): “We undertook a prospective single-center cross-validation study during one year.” It is advised to describe the periods and dates of participant recruitment and data collection if the information is available.

2.      Methods (Participants, Line 109 on Page 3): The study included 94 adults with ID. The methods of recruitment (for instance, referral) were unclear. In addition, it is unclear how many subjects were screened for eligibility, how many screened subjects were eligible, and how many eligible subjects agreed to enter the study. The authors may consider a more detailed description.

3.      Results (Table 1): The demographic characteristics according to the groups were presented. It is unclear why the percentage of males and females in each group did not add up to one.  

4.      Results (Tables 1-5): “*p < 0.05, **p < 0.01, *** p < 0.001 according Kruskal-Wallis with Bonferroni correction.” What tests were the p-values referred to? Were they referred to the comparisons with CN or else? A clarification is suggested.

5.      Results (Line 281-283 on page 9): “The PMIS-ID was applied again in a convenience subsample of 20 participants within --.” The authors described how reliability was determined. Please consider moving this description to the Methods.

6.      Results (Discriminative validity, Line 301- 310 on Pages 9-10): The results of ROC analysis in participants with moderate ID were presented. However, the analytic results in total participants and those with mild ID would also be informative. The authors may consider presenting these additional findings (for instance, in the supplement).

Reviewer 2 Report

The abstract is informative of the content of the paper. Given the results, I suggest authors refer in the title to adults with moderate intellectual disability.

Page 2. lines 73-78. Memory, both verbal and visual,  is mentioned as important together with executive functioning. But then the focus is moved to memory and the need is mentioned for testing: “is evident that memory declines across the AD continuum”: same holds for executive functioning too. “ So, there is a need to develop sensitive and screening tests for daily clinical practice to detect memory changes associated with MCI or AD in this population”: the need to develop sensitive screening tests for daily clinical practice to detect memory changes and not the executive processes are not clear and motivated in detail.

Page 3, lines 112-119. In reporting information on diagnosis, no mention is made of either quantitative or qualitative tools used nor to the degree of change concerning the previous functioning. Please, briefly add this information.  No mention is given to the etiology of the participants’ ID. Please, briefly add this information

Page 3, line 120: Normative data mentioned here are referred to as ID population. Please specify here and later in the paper ( page 10l for example)

Page 3, line 162  The PMIS-ID is described here but later in more detail again (173). I suggest authors modify the organization of the paragraph or just mention the PMID-ID and inform readers that it will be described later.

Page 9. Please motivate the  shift to the use of  parametric tests, such as  for the  correlation  

Page 13, line 463.  The authors only at this point mention the relevance of this study for adults with moderate ID. Actually, in several points of the discussion, the authors should comment on the results making specific reference to individuals with moderate ID.  The highest use of the tool is specifically in working with them.

So, they should also comment and anticipate possible developments to make it more useful for individuals with mild ID.
